# Stochastic Normalizing Flows

**Hao Wu**
Tongji University
Shanghai, P.R. China
wwtian@gmail.com

**Jonas Köhler**
FU Berlin
Berlin, Germany
jonas.koehler@fu-berlin.de

**Frank Noé**
FU Berlin
Berlin, Germany
frank.noe@fu-berlin.de

## Abstract

The sampling of probability distributions specified up to a normalization constant
is an important problem in both machine learning and statistical mechanics. While
classical stochastic sampling methods such as Markov Chain Monte Carlo (MCMC)
or Langevin Dynamics (LD) can suffer from slow mixing times there is a growing
interest in using normalizing flows in order to learn the transformation of a simple
prior distribution to the given target distribution. Here we propose a generalized
and combined approach to sample target densities: Stochastic Normalizing Flows
(SNF) – an arbitrary sequence of deterministic invertible functions and stochastic
sampling blocks. We show that stochasticity overcomes expressivity limitations
of normalizing flows resulting from the invertibility constraint, whereas trainable
transformations between sampling steps improve efficiency of pure MCMC/LD
along the flow. By invoking ideas from non-equilibrium statistical mechanics
we derive an efficient training procedure by which both the sampler's and the
flow's parameters can be optimized end-to-end, and by which we can compute
exact importance weights without having to marginalize out the randomness of the
stochastic blocks. We illustrate the representational power, sampling efficiency and
asymptotic correctness of SNFs on several benchmarks including applications to
sampling molecular systems in equilibrium.

## 1 Introduction

A common problem in machine learning and statistics with important applications in physics is the
generation of asymptotically unbiased samples from a target distribution defined up to a normalization
constant by means of an energy model $u(\mathbf{x})$:

$$\mu_X(\mathbf{x}) \propto \exp(-u(\mathbf{x})). \tag{1}$$

Sampling of such unnormalized distributions is often done with Markov Chain Monte Carlo (MCMC)
or other stochastic sampling methods [13]. This approach is asymptotically unbiased, but suffers
from the sampling problem: without knowing efficient moves, MCMC approaches may get stuck in
local energy minima for a long time and fail to converge in practice.

Normalizing flows (NFs) [41, 40, 5, 35, 6, 33] combined with importance sampling methods are
an alternative approach that enjoys growing interest in molecular and material sciences and nuclear
physics [28, 25, 32, 22, 1, 30]. NFs are learnable invertible functions, usually represented by a neural
network, pushing forward a probability density over a latent or "prior" space $Z$ towards the target
space $X$. Utilizing the change of variable rule these models provide exact densities of generated
samples allowing them to be trained by either maximizing the likelihood on data (ML) or minimizing
the Kullback-Leibler divergence (KL) towards a target distribution.

Let $F_{ZX}$ be such a map and its inverse $F_{XZ} = F_{ZX}^{-1}$. We can consider it as composition of $T$
invertible transformation layers $F_0, ..., F_T$ with intermediate states $\mathbf{y}_t$ given by:

$$\mathbf{y}_{t+1} = F_t(\mathbf{y}_t) \qquad \mathbf{y}_t = F_t^{-1}(\mathbf{y}_{t+1}) \tag{2}$$

By calling the samples in $Z$ and $X$ also $\mathbf{z}$ and $\mathbf{x}$, respectively, the flow structure is as follows:

$$\mathbf{z} = \mathbf{y}_0 \overset{F_0}{\underset{F_0^{-1}}{\rightleftarrows}} \mathbf{y}_1 \rightleftarrows \cdots \rightleftarrows \mathbf{y}_{T-1} \overset{F_{T-1}}{\underset{F_{T-1}^{-1}}{\rightleftarrows}} \mathbf{y}_T = \mathbf{x} \tag{3}$$

We suppose each transformation layer is differentiable with a Jacobian determinant $|\det \mathbf{J}_t(\mathbf{y})|$. This allows to apply the *change of variable* rule:

$$p_{t+1}(\mathbf{y}_{t+1}) = p_{t+1}\left(F_t(\mathbf{y}_t)\right) = p_t(\mathbf{y}_t)\left|\det \mathbf{J}_t(\mathbf{y}_t)\right|^{-1}. \tag{4}$$

As we often work with log-densities, we abbreviate the log Jacobian determinant as:

$$\Delta S_t = \log\left|\det \mathbf{J}_t(\mathbf{y})\right|. \tag{5}$$

The log Jacobian determinant of the entire flow is defined by $\Delta S_{ZX} = \sum_t \Delta S_t(\mathbf{y}_t)$ and correspondingly $\Delta S_{XZ}$ for the inverse flow.

**Unbiased sampling with Boltzmann Generators.** Unbiased sampling is particularly important for applications in physics and chemistry where unbiased expectation values are required [25, 32, 1, 30]. A Boltzmann generator [32] utilizing NFs achieves this by (i) generating one-shot samples $\mathbf{x} \sim p_X(\mathbf{x})$ from the flow and (ii) using a reweighing/resampling procedure respecting weights

$$w(\mathbf{x}) = \frac{\mu_X(\mathbf{x})}{p_X(\mathbf{x})} \propto \exp\left(-u_X(\mathbf{x}) + u_Z(\mathbf{z}) + \Delta S_{ZX}(\mathbf{z})\right), \tag{6}$$

turning these one-shot samples into asymptotically unbiased samples. Reweighing/resampling methods utilized in this context are e.g. *Importance Sampling* [28, 32] or *Neural MCMC* [25, 1, 30].

**Training NFs.** NFs are trained in either "forward" or "reverse" mode, e.g.:

1. Density estimation – given data samples $\mathbf{x}$, train the flow such that the back-transformed samples $\mathbf{z} = F_{XZ}(\mathbf{x})$ follow a latent distribution $\mu_Z(\mathbf{z})$, e.g. $\mu_Z(\mathbf{z}) = \mathcal{N}(\mathbf{0}, \mathbf{I})$. This is done by maximizing the likelihood – equivalent to minimizing the KL divergence $KL\left[\mu_X \| p_X\right]$.

2. Sampling of a given target density $\mu_X(\mathbf{x})$ – sample from the simple distribution $\mu_Z(\mathbf{z})$ and minimize a divergence between the distribution generated by the forward-transformation $\mathbf{x} = F_{XZ}(\mathbf{z})$ and $\mu_X(\mathbf{x})$. A common choice is the reverse KL divergence $KL\left[p_X \| \mu_X\right]$.

We will use densities interchangeably with energies, defined by the negative logarithm of the density. The exact prior and target distributions are:

$$\mu_Z(\mathbf{z}) = Z_Z^{-1}\exp(-u_Z(\mathbf{z})) \qquad \mu_X(\mathbf{x}) = Z_X^{-1}\exp(-u_X(\mathbf{x})) \tag{7}$$

with generally unknown normalization constants $Z_Z$ and $Z_X$. As can be shown (Suppl. Material Sec. 1) minimizing $KL\left[p_X \| \mu_X\right]$ or $KL\left[\mu_X \| p_X\right]$ corresponds to maximizing the forward or backward weights of samples drawn from $p_X$ or $\mu_X$, respectively.

**Topological problems of NFs.** A major caveat of sampling with exactly invertible functions for physical problems are topological constraints. While these can be strong manifold results, e.g., if the sample space is restricted to a non-trivial Lie group [11, 12], another practical problem are induced Bi-Lipschitz constraints resulting from mapping uni-modal base distributions onto well-separated multi-modal target distributions[4]. For example, when trying to map a unimodal Gaussian distribution to a bimodal distribution with affine coupling layers, a connection between the modes remains (Fig. 1a). This representational insufficiency poses serious problems during optimization – in the bimodal distribution example, the connection between the density modes seems largely determined by the initialization and does not move during optimization, leading to very different results in multiple runs (Suppl. Material, Fig. S1). More powerful coupling layers, e.g., [9], can mitigate this effect. Yet, as they are still diffeomorphic, strong Bi-Lipschitz requirements can make optimization difficult. This problem can be resolved when relaxing bijectivity of the flow by adding noise as we show in our results. Other proposed solutions are real-and-discrete mixtures of flows [7] or augmentation of the bases space [8, 18] at the cost of losing asymptotically unbiased sampling.

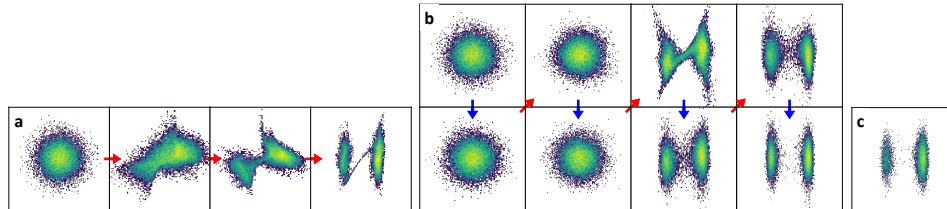

Figure 1: **Deterministic versus stochastic normalizing flow for the double well**. Red arrows indicate deterministic transformations, blue arrows indicate stochastic dynamics. **a**) 3 RealNVP blocks (2 layers each). **b**) Same with 20 BD steps before or after RealNVP blocks. **c**) Unbiased sample from true distribution.

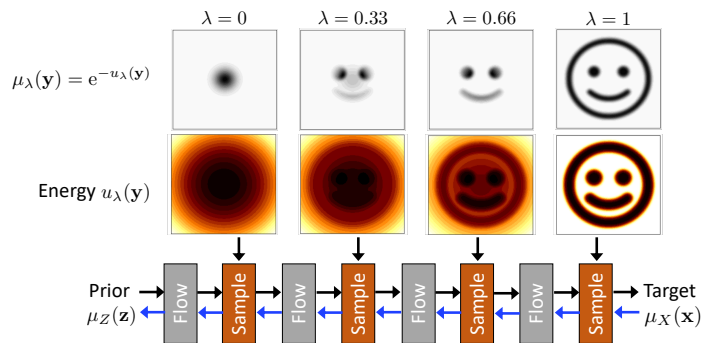

Figure 2: **Schematic for Stochastic Normalizing Flow (SNF)**. An SNF transforms a tractable prior $\mu_Z(\mathbf{z}) \propto \exp(-u_0(\mathbf{z}))$ to a complicated target distribution $\mu_X(\mathbf{x}) \propto \exp(-u_1(\mathbf{x}))$ by a sequence of deterministic invertible transformations (flows, grey boxes) and stochastic dynamics (sample, ochre) that sample with respect to a guiding potential $u_\lambda(\mathbf{x})$. SNFs can be trained and run in forward mode (black) and reverse mode (blue).

**Contributions.**    We show that NFs can be interwoven with stochastic sampling blocks into arbitrary sequences, that together overcome topological constraints and improve expressivity over deterministic flow architectures (Fig. 1a, b). Furthermore, NSFs have improved sampling efficiency over pure stochastic sampling as the flow's and sampler's parameters can be optimized jointly.

Our main result is that NSFs can be trained in a similar fashion as NFs and exact importance weights for each sample ending in $\mathbf{x}$ can be computed, facilitating asymptotically unbiased sampling from the target density. The approach avoids explicitly computing $p_X(\mathbf{x})$ which would require solving the intractable integral over all stochastic paths ending in $\mathbf{x}$.

We apply the model to the recently introduced problem of asymptotically unbiased sampling of molecular structures with flows [32] and show that it significantly improves sampling the multi-modal torsion angle distributions which are the relevant degrees of freedom in the system. We further show the advantage of the method over pure flow-based sampling / MCMC by quantitative comparison on benchmark data sets and on sampling from a VAE's posterior distribution.

**Code** is available at `github.com/noegroup/stochastic_normalizing_flows`

## 2   Stochastic normalizing flows

A SNF is a sequence of $T$ stochastic and deterministic transformations. We sample $\mathbf{z} = \mathbf{y}_0$ from the prior $\mu_Z$, and generate a forward path $(\mathbf{y}_1, \ldots, \mathbf{y}_T)$ resulting in a proposal $\mathbf{y}_T$ (Fig. 2). Correspondingly, latent space samples can be generated by starting from a sample $\mathbf{x} = \mathbf{y}_T$ and invoking the backward path $(\mathbf{y}_{T-1}, \ldots, \mathbf{y}_0)$. The conditional forward / backward path probabilities are

$$\mathbb{P}_f(\mathbf{z}{=}\mathbf{y}_0 \to \mathbf{y}_T{=}\mathbf{x}) = \prod_{t=0}^{T-1} q_t(\mathbf{y}_t \to \mathbf{y}_{t+1}), \quad \mathbb{P}_b(\mathbf{x}{=}\mathbf{y}_T \to \mathbf{y}_0{=}\mathbf{z}) = \prod_{t=0}^{T-1} \tilde{q}_t(\mathbf{y}_{t+1} \to \mathbf{y}_t) \quad (8)$$

where

$$, \mathbf{y}_{t+1}|\mathbf{y}_t \sim q_t(\mathbf{y}_t \to \mathbf{y}_{t+1}) \qquad \mathbf{y}_t|\mathbf{y}_{t+1} \sim \tilde{q}_t(\mathbf{y}_{t+1} \to \mathbf{y}_t) \tag{9}$$

denote the forward / backward sampling density at step $t$ respectively. If step $t$ is a deterministic transformation $F_t$ this simplifies as

$$\mathbf{y}_{t+1} \sim \delta\left(\mathbf{y}_{t+1} - F_t(\mathbf{y}_t)\right), \qquad \mathbf{y}_t \sim \delta\left(\mathbf{y}_t - F_t^{-1}(\mathbf{y}_{t+1})\right).$$

In contrast to NFs, the probability that an SNF generates a sample $\mathbf{x}$ cannot be computed by Eq. (4) but instead involves an integral over all paths that end in $\mathbf{x}$:

$$p_X(\mathbf{x}) = \int \mu_Z(\mathbf{y}_0)\mathbb{P}_f(\mathbf{y}_0 \to \mathbf{y}_T)\,\mathrm{d}\mathbf{y}_0 \cdots \mathrm{d}\mathbf{y}_{T-1}. \tag{10}$$

This integral is generally intractable, thus a feasible training method must avoid using Eq. (10). Following [31], we can draw samples $\mathbf{x} \sim \mu_X(\mathbf{x})$ by running Metropolis-Hastings moves in the path-space of $(\mathbf{z} = \mathbf{y}_0, ..., \mathbf{y}_T = \mathbf{x})$ if we select the backward path probability $\mu_X(\mathbf{x})\mathbb{P}_b(\mathbf{x} \to \mathbf{z})$ as the target distribution and the forward path probability $\mu_Z(\mathbf{z})\mathbb{P}_f(\mathbf{z} \to \mathbf{x})$ as the proposal density. Since we sample paths independently, it is simpler to assign an unnormalized importance weight proportional to the acceptance ratio to each sample path from $\mathbf{z} = \mathbf{y}_0$ to $\mathbf{x} = \mathbf{y}_T$:

$$w(\mathbf{z} \to \mathbf{x}) = \exp\left(-u_X(\mathbf{x}) + u_Z(\mathbf{z}) + \sum_t \Delta S_t(\mathbf{y}_t)\right) \propto \frac{\mu_X(\mathbf{x})\mathbb{P}_b(\mathbf{x} \to \mathbf{z})}{\mu_Z(\mathbf{z})\mathbb{P}_f(\mathbf{z} \to \mathbf{x})}, \tag{11}$$

where

$$\Delta S_t = \log \frac{\tilde{q}_t(\mathbf{y}_{t+1} \to \mathbf{y}_t)}{q_t(\mathbf{y}_t \to \mathbf{y}_{t+1})} \tag{12}$$

denotes the forward-backward probability ratio of step $t$, and corresponds to the usual change of variable formula in NF for deterministic transformation steps (Suppl. Material Sec. 3). These weights allow asymptotically unbiased sampling and training of SNFs while avoiding Eq. (10). By changing denominator and numerator in (11) we can alternatively obtain the backward weights $w(\mathbf{x} \to \mathbf{z})$.

**SNF training.** As in NFs, the parameters of a SNF can be optimized by minimizing the Kullback-Leibler divergence between the forward and backward path probabilities, or alternatively maximizing forward and backward path weights as long as we can compute $\Delta S_t$ (Suppl. Material Sec 1):

$$J_{\mathrm{KL}} = \mathbb{E}_{\mu_Z(\mathbf{z})\mathbb{P}_f(\mathbf{z}\to\mathbf{x})} \left[-\log w(\mathbf{z} \to \mathbf{x})\right] = \mathrm{KL}\left(\mu_Z(\mathbf{z})\mathbb{P}_f(\mathbf{z} \to \mathbf{x})||\mu_X(\mathbf{x})\mathbb{P}_b(\mathbf{x} \to \mathbf{z})\right) + \mathrm{const}. \tag{13}$$

In the ideal case of $J_{\mathrm{KL}} = 0$, all paths have the same weight $w(\mathbf{z} \to \mathbf{x}) = 1$ and the independent and identically distributed sampling of $\mu_X$ can be achieved. Accordingly, we can maximize the likelihood of the generating process on data drawn from $\mu_X$ by minimizing:

$$J_{\mathrm{ML}} = \mathbb{E}_{\mu_X(\mathbf{x})\mathbb{P}_b(\mathbf{x}\to\mathbf{z})} \left[-\log w(\mathbf{x} \to \mathbf{z})\right] = \mathrm{KL}\left(\mu_X(\mathbf{x})\mathbb{P}_b(\mathbf{x} \to \mathbf{z})||\mu_Z(\mathbf{z})\mathbb{P}_f(\mathbf{z} \to \mathbf{x})\right) + \mathrm{const}. \tag{14}$$

**Variational bound.** Minimization of the reverse path divergence $J_{KL}$ minimizes an upper bound on the reverse KL divergence between the marginal distributions:

$$\mathrm{KL}\left(p_X(\mathbf{x}) \parallel \mu_X(\mathbf{x})\right) \leq \mathrm{KL}\left(\mu_Z(\mathbf{z})\mathbb{P}_f(\mathbf{z} \to \mathbf{x}) \parallel \mu_X(\mathbf{x})\mathbb{P}_b(\mathbf{x} \to \mathbf{z})\right) \tag{15}$$

And the same relationship exists between the forward path divergence $J_{ML}$ and the forward KL divergence. While invoking this variational approximation precludes us from explicitly computing $p_X(\mathbf{x})$ and $\mathrm{KL}\left(p_X(\mathbf{x}) \parallel \mu_X(\mathbf{x})\right)$, we can still generate asymptotically unbiased samples from the target density $\mu_X$, unlike in variational inference.

**Asymptotically unbiased sampling.** As stated in the theorem below (Proof in Suppl. Material. Sec. 2), SNFs are Boltzmann Generators: We can generate asymptotically unbiased samples of $\mathbf{x} \sim \mu_X(\mathbf{x})$ by performing importance sampling or Neural MCMC using the path weight $w(\mathbf{z}_k \to \mathbf{x}_k)$ of each path sample $k$.

**Theorem 1.** *Let $O$ be a function over $X$. An asymptotically unbiased estimator is given by*

$$\mathbb{E}_{\mathbf{x}\sim\mu_X}\left[O(\mathbf{x})\right] \approx \frac{\sum_k w(\mathbf{z}_k \to \mathbf{x}_k)\,O(\mathbf{x}_k)}{\sum_k w(\mathbf{z}_k \to \mathbf{x}_k)}, \tag{16}$$

*if paths are drawn from the forward path distribution $\mu_Z(\mathbf{z})\mathbb{P}_f(\mathbf{z} \to \mathbf{x})$.*

## 3 Implementing SNFs via Annealed Importance Sampling

In this paper we focus on the use of SNFs as samplers of $\mu_X(\mathbf{x})$ for problems where the target energy $u_X(\mathbf{x})$ is known, defining the target density up to a constant, and provide an implementation of stochastic blocks via MCMC / LD. These blocks make local stochastic updates of the current state $\mathbf{y}$ with respect to some potential $u_\lambda(\mathbf{y})$ such that they will asymptotically sample from $\mu_\lambda(\mathbf{y}) \propto \exp(-u_\lambda(\mathbf{y}))$. While such potentials $u_\lambda(\mathbf{y})$ could be learned, a straightforward strategy is to interpolate between prior and target potentials

$$u_\lambda(\mathbf{y}) = (1 - \lambda)u_Z(\mathbf{y}) + \lambda u_X(\mathbf{y}), \tag{17}$$

similarly as it is done in *annealed importance sampling* [29]. Our implementation for SNFs is thus as follows: deterministic flow layers in-between only have to approximate the partial density transformation between adjacent $\lambda$ steps while the stochastic blocks anneal with respect to the given intermediate potential $u_\lambda$. The parameter $\lambda$ could again be learned – in this paper we simply choose a linear interpolation along the SNF layers: $\lambda = t/T$.

**Langevin dynamics.** Overdamped Langevin dynamics, also known as Brownian dynamics, using an Euler discretization with time step $\Delta t$, are given by [10]:

$$\mathbf{y}_{t+1} = \mathbf{y}_t - \epsilon_t \nabla u_\lambda(\mathbf{y}_t) + \sqrt{2\epsilon_t/\beta}\boldsymbol{\eta}_t, \tag{18}$$

where $\boldsymbol{\eta}_t \sim \mathcal{N}(0, \mathbf{I})$ is Gaussian noise. In physical systems, the constant $\epsilon_t$ has the form $\epsilon_t = \Delta t/\gamma m$ with time step $\Delta t$, friction coefficient $\gamma$ and mass $m$, and $\beta$ is the inverse temperature (here set to 1). The backward step $\mathbf{y}_{t+1} \to \mathbf{y}_t$ is realized under these dynamics with the backward noise realization (Suppl. Material Sec. 4 and [31]):

$$\tilde{\boldsymbol{\eta}}_t = \sqrt{\frac{\beta\epsilon_t}{2}}\left[\nabla u_\lambda(\mathbf{y}_t) + \nabla u_\lambda(\mathbf{y}_{t+1})\right] - \boldsymbol{\eta}_t. \tag{19}$$

The log path probability ratio is (Suppl. Material Sec. 4):

$$\Delta S_t = -\frac{1}{2}\left(\|\tilde{\boldsymbol{\eta}}_t\|^2 - \|\boldsymbol{\eta}_t\|^2\right). \tag{20}$$

We also give the results for non-overdamped Langevin dynamics in Suppl. Material. Sec. 5.

**Markov Chain Monte Carlo.** Consider MCMC methods with a proposal density $q_t$ that satisfies the detailed balance condition w.r.t. the interpolated density $\mu_\lambda(\mathbf{y}) \propto \exp(-u_\lambda(\mathbf{y}))$:

$$\exp(-u_\lambda(\mathbf{y}_t))q_t(\mathbf{y}_t \to \mathbf{y}_{t+1}) = \exp(-u_\lambda(\mathbf{y}_{t+1}))q_t(\mathbf{y}_{t+1} \to \mathbf{y}_t) \tag{21}$$

We show that for all $q_t$ satisfying (21), including Metropolis-Hastings and Hamiltonian MC moves, the log path probability ratio is (Suppl. Material Sec. 6 and 7):

$$\Delta S_t = u_\lambda(\mathbf{y}_{t+1}) - u_\lambda(\mathbf{y}_t), \tag{22}$$

if the backward sampling density satisfies $\tilde{q}_t = q_t$.

## 4 Results

**Representational power versus sampling efficiency.** We first illustrate that SNFs can break topological constraints and improve the representational power of deterministic normalizing flows at a given network size and at the same time beat direct MCMC in terms of sampling efficiency. To this end we use images to define complex two-dimensional densities (Fig. 3a-c, "Exact") as target densities $\mu_X(\mathbf{x})$ to be sampled. Note that a benchmark aiming at generating high-quality images would instead represent the image as a high-dimensional pixel array. We compare three types of flows with 5 blocks each trained by samples from the exact density (details in Suppl. Material Sec. 9):

1. Normalizing flow with 2 swapped coupling layers (RealNVP or neural spline flow) per block
2. Non-trainable stochastic flow with 10 Metropolis MC steps per block
3. SNF with both, 2 swapped coupling layers and 10 Metropolis MC steps per block.

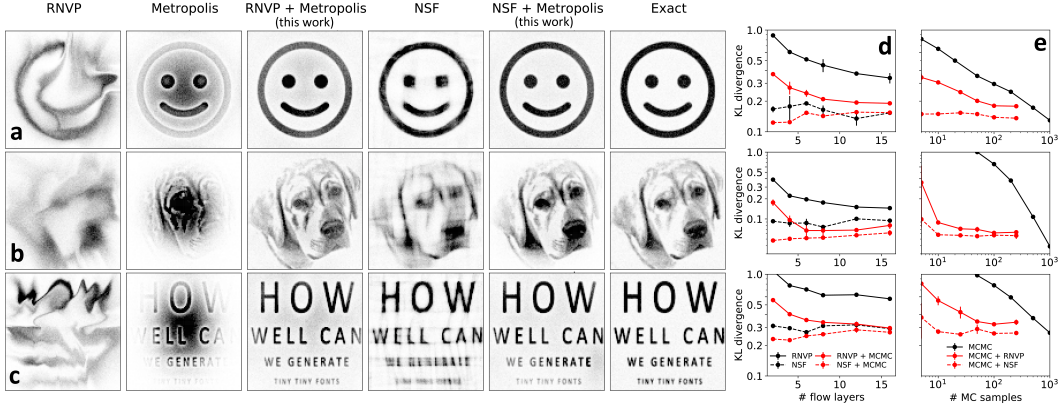

Figure 3: **Sampling of two-dimensional densities**. **a-c)** Sampling of smiley, dog and text densities with different methods. Columns: (1) Normalizing Flow with RealNVP layers, (2) Metropolis MC sampling, (3) Stochastic Normalizing Flow combining (1+2), (4) neural spline flow (NSF), (5) Stochastic Normalizing Flow combining (1+4), (6) Unbiased sample from exact density. **d-e)** Compare representative power and statistical efficiency of different flow methods by showing KL divergence (mean and standard deviation over 3 training runs) between flow samples and true density for the three images from Fig. 3. **d)** Comparison of deterministic flows (black) and SNF (red) as a function of the number of RealNVP or Neural Spline Flow transformations. Total number of MC steps in SNF is fixed to 50. **e)** Comparison of pure Metropolis MC (black) and SNF (red, solid line RealNVP, dashed line Neural spline flow) as a function of the number of MC steps. Total number of RealNVP or NSF transformations in SNF is fixed to 10.

The pure Metropolis MC flow suffers from sampling problems – density is still concentrated in the image center from the prior. Many more MC steps would be needed to converge to the exact density (see below). The RealNVP normalizing flow architecture [6] has limited representational power, resulting in a "smeared out" image that does not resolve detailed structures (Fig. 3a-c, RNVP). As expected, neural spline flows perform significantly better on the 2D-images than RealNVP flows, but at the chosen network architecture their ability to resolve fine details and round shapes is still limite (See dog and small text in Fig. 3c, NSF). Note that the representational power for all flow architectures tend to increase with depth - here we compare the performance of different architectures at fixed depth and similar computational cost.

In contrast, SNFs achieve high-quality approximations although they simply combine the same deterministic and stochastic flow components that fail individually in the SNF learning framework (Fig. 3a-c, RNVP+Metropolis and NSF+Metropolis). This indicates that the SNF succeeds in performing the large-scale probability mass transport with the trainable flow layers and sampling the details with Metropolis MC.

Fig. 3d-e quantifies these impressions by computing the KL divergence between generated densities $p_X(\mathbf{x})$ and exact densities $\mu_X(\mathbf{x})$. Both normalizing flows and SNFs improve with greater depth, but SNFs achieve significantly lower KL divergence at a fixed network depth (Fig. 3d). Note that both RealNVP and NSFs improve significantly when stochasticty is added.

Moreover, SNFs have higher statistical efficiency than pure Metropolis MC flows. Depending on the example and flow architecture, 1-2 orders of magnitude more Metropolis MC steps are needed to achieve similar KL divergence as with an SNF. This demonstrates that the large-scale probability transport learned by the trainable deterministic flow blocks in SNFs significantly helps with the sampling.

Importantly, adding stochasticity is very inexpensive. Although every MCMC or Langevin integration step adds a neural network layer, these layers are very lightweighted, and have only linear computational complexity in the number of dimensions. As an example, for our SNF implementation of the examples in Fig. 3 we can add 10-20 stochastic layers to each trainable normalizing flow layer before the computational cost increases by a factor of 2 (Suppl. Material Fig. S2).

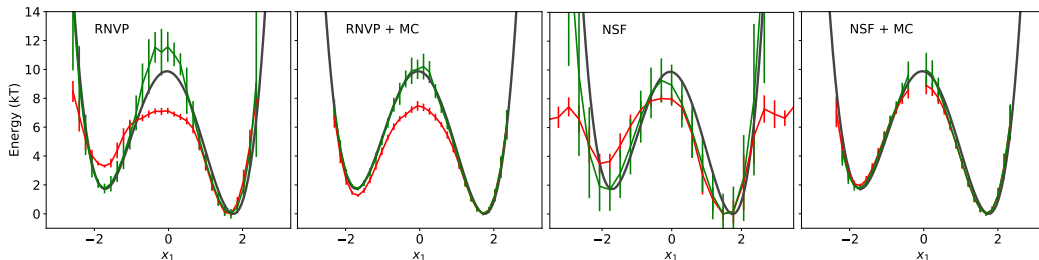

Figure 4: **Reweighting results for the double well potential** (see also Fig. 1). Free energy along $x_1$ (negative log of marginal density) for deterministic normalizing flows (RNVP, NSF) and SNFs (RNVP+MC, NSF+MC). Black: exact energy, red: energy of proposal density $p_X(\mathbf{x})$, green: reweighted energy using importance sampling.

**SNFs as asymptotically unbiased samplers.** We demonstrate that SNFs can be used as Boltzmann Generators, i.e., to sample target densities without asymptotic bias by revisiting the double-well example (Fig. 1). Fig. 4 (black) shows the free energies (negative marginal density) along the double-well coordinate $x_1$. Flows with 3 coupling layer blocks (RealNVP or neural spline flow) are trained summing forward and reverse KL divergence as a joint loss using either data from a biased distribution, or with the unbiased distribution (Details in Suppl. Material Sec. 9). Due to limitations in representational power the generation probability $p_X(\mathbf{x})$ will be biased – even when explicitly minimizing the KL divergence w.r.t. the true unbiased distribution in the joint loss. By relying on importance sampling we can turn the flows into Boltzmann Generators [32] in order to obtain unbiased estimates. Indeed all generator densities $p_X(\mathbf{x})$ can be reweighted to an estimate of the unbiased density $\mu_X(\mathbf{x})$ whose free energies are within statistical error of the exact result (Fig. 4, red and green).

We inspect the bias, i.e. the error of the mean estimator, and the statistical uncertainty ($\sqrt{\text{var}}$) of the free energy in $x_1 \in \{-2.5, 2.5\}$ with and without reweighting using a fixed number of samples (100,000). Using SNFs with Metropolis MC steps, both biases and uncertainties are reduced by half compared to purely deterministic flows (Table 1). Note that neural spline flows perform better than RealNVP without reweighting, but significantly worse with reweighting - presumably because the sharper features representable by splines can be detrimental for reweighting weights. With stochastic layers, both RealNVP and neural spline flows perform approximately equally well.

The differences between multiple runs (see standard deviations of the uncertainty estimate) also reduce significantly, i.e. SNF results are more reproducible than RealNVP flows, confirming that the training problems caused by the density connection between both modes (Fig. 1, Suppl. Material Fig. S1) can be reduced. Moreover, the sampling performance of SNF can be further improved by optimizing MC step sizes based on loss functions $J_{KL}$ and $J_{ML}$ (Suppl. Material Table S1).

Reweighting reduces the bias at the expense of a higher variance. Especially in physics applications, a small or asymptotically zero bias is often very important, and the variance can be reduced by generating more samples from the trained flow, which is relatively cheap and parallel.

Table 1: **Unbiased sampling for double well potential:** mean uncertainty of the reweighted energy along $x_1$ averaged over 10 independent runs ($\pm$ standard deviation).

|  | not reweighted | | | reweighted | | |
|---|---|---|---|---|---|---|
|  | bias | $\sqrt{\text{var}}$ | $\sqrt{\text{bias}^2+\text{var}}$ | bias | $\sqrt{\text{var}}$ | $\sqrt{\text{bias}^2+\text{var}}$ |
| RNVP | $1.4 \pm 0.6$ | $0.4 \pm 0.1$ | $1.5 \pm 0.5$ | $0.3 \pm 0.2$ | $1.1 \pm 0.4$ | $1.2 \pm 0.4$ |
| **RNVP + MC** | $1.5 \pm 0.2$ | $0.3 \pm 0.1$ | $1.5 \pm 0.2$ | $\mathbf{0.2 \pm 0.1}$ | $\mathbf{0.6 \pm 0.1}$ | $\mathbf{0.6 \pm 0.1}$ |
| NSF | $0.8 \pm 0.4$ | $1.0 \pm 0.2$ | $1.3 \pm 0.3$ | $0.6 \pm 0.2$ | $2.1 \pm 0.4$ | $2.2 \pm 0.5$ |
| **NSF + MC** | $0.4 \pm 0.3$ | $0.5 \pm 0.1$ | $0.7 \pm 0.2$ | $\mathbf{0.1 \pm 0.1}$ | $\mathbf{0.6 \pm 0.2}$ | $\mathbf{0.6 \pm 0.2}$ |

**Alanine dipeptide.** We further evaluate SNFs on density estimation and sampling of molecular structures from a simulation of the alanine dipeptide molecule in vacuum (Fig. 5). The molecule has 66 dimensions in $\mathbf{x}$, and we augment it with 66 auxiliary dimensions in a second channel $\mathbf{v}$, similar

to "velocities" in a Hamiltonian flow framework [42], resulting in 132 dimensions total. The target density is given by $\mu_X(\mathbf{x}, \mathbf{v}) = \exp\left(-u(\mathbf{x}) - \frac{1}{2}\|\mathbf{v}\|^2\right)$, where $u(\mathbf{x})$ is the potential energy of the molecule and $\frac{1}{2}\|\mathbf{v}\|^2$ is the kinetic energy term. $\mu_Z$ is an isotropic Gaussian normal distribution in all dimensions. We utilize the invertible coordinate transformation layer introduced in [32] in order to transform $\mathbf{x}$ into normalized bond, angle and torsion coordinates. RealNVP transformations act between the $\mathbf{x}$ and $\mathbf{v}$ variable groups Details in Suppl. Material Sec. 9).

We compare deterministic normalizing flows using 5 blocks of 2 RealNVP layers with SNFs that additionally use 20 Metropolis MC steps in each block totalling up to 100 MCMC steps in one forward pass. Fig. 5a shows random structures sampled by the trained SNF. Fig. 5b shows marginal densities in all five multimodal torsion angles (backbone angles $\phi$, $\psi$ and methyl rotation angles $\gamma_1$, $\gamma_2$, $\gamma_3$). While the RealNVP networks that are state of the art for this problem miss many of the modes, the SNF resolves the multimodal structure and approximates the target distribution better, as quantified in the KL divergence between the generated and target marginal distributions (Table 2).

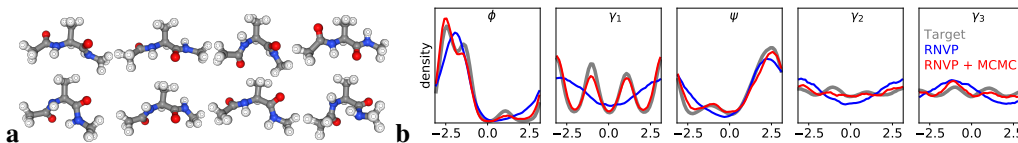

Figure 5: **Alanine dipeptide** sampled with deterministic normalizing flows and stochastic normalizing flows. **a**) One-shot SNF samples of alanine dipeptide structures. **b**) Energy (negative logarithm) of marginal densities in 5 unimodal torsion angles (top) and all 5 multimodal torsion angles (bottom).

Table 2: **Alanine dipeptide**: KL-divergences of RNVP flow and SNF (RNVP+MCMC) between generated and target distributions for all multimodal torsion angles. Mean and standard deviation from 3 independent runs.

| KL-div. | $\phi$ | $\gamma_1$ | $\psi$ | $\gamma_2$ | $\gamma_3$ |
|---|---|---|---|---|---|
| RNVP | 1.69±0.03 | 3.82±0.01 | 0.98±0.03 | 0.79±0.03 | 0.79±0.09 |
| SNF | **0.36 ± 0.05** | **0.21** ±0.01 | **0.27** ±0.03 | **0.12** ±0.02 | **0.15** ±0.04 |

**Variational Inference.** Finally, we use normalizing flows to model the latent space distribution of a variational autoencoder (VAE) , as suggested in [35]. Table 3 shows results for the variational bound and the log likelihood on the test set for MNIST [23] and Fashion-MNIST [44]. For a 50-dimensional latent space we compare a six-layer RNVP to MCMC using overdamped Langevin dynamics as proposal (MCMC) and a SNF combining both (RNVP+MCMC). Both sampling and the deterministic flow improve over a naive VAE using a reparameterized diagonal Gaussian variational posterior distribution, while the SNF outperforms both, RNVP and MCMC. See Suppl. Material Sec. 8 for details.

Table 3: **Variational inference using VAEs with stochastic normalizing flows**: $J_{KL}$: variational bound of the KL-divergence computed during training. NLL: negative log likelihood of test set.

| | MNIST | | Fashion-MNIST | |
|---|---|---|---|---|
| | $J_{KL}$ | NLL | $J_{KL}$ | NLL |
| Naive (Gaussian) | 108.4±24.3 | 98.1±4.2 | 241.3±7.4 | 238.0±2.9 |
| RNVP | 91.8±0.4 | 87.0±0.2 | 233.7±0.1 | 231.4±0.2 |
| MCMC | 102.1±8.0 | 96.2±1.9 | 234.7±0.4 | 235.2±2.4 |
| SNF | **89.7±0.1** | **86.8±0.1** | **232.4±0.2** | **230.9±0.2** |

# 5 Related work

The cornerstone of our work is nonequilibrium statistical mechanics. Particularly important is Nonequilibrium Candidate Monte Carlo (NCMC) [31], which provides the theoretical framework to compute SNF path likelihood ratios. However, NCMC is for fixed deterministic and stochastic protocols, while we generalize this into a generative model by substituting fixed protocols with trainable layers and deriving an unbiased optimization procedure.

Neural stochastic differential equations learn optimal parameters of designed stochastic processes from observations along the path [43, 19, 27, 26], but are not designed for marginal density estimation or asymptotically unbiased sampling. It has been demonstrated that combining learnable proposals/transformations with stochastic sampling techniques can improve expressiveness of the proposals [36, 24, 39, 17, 16]. Yet, these contributions do not provide an exact reweighing scheme based on a tractable model likelihood and do not provide efficient algorithms to optimize arbitrary sequences of transformation or sampling steps end-to-end efficiently. These methods can be seen as instances of SNFs with specific choice of deterministic transformations and / or stochastic blocks and model-specific optimizations - see Suppl. Material Table S2) for a categorization. While our experiments focus on nontrainable stochastic blocks, the proposal densities of MC steps can also be optimized within the framework of SNFs as shown in Suppl. Material Table S1.

An important aspect of SNFs compared to trainable Monte-Carlo kernels such as A-NICE-MC [39] is the use of detailed balance (DB). While Monte-Carlo frameworks are usually designed to use DB in each step, SNFs rely on path-based detailed balance between the prior and the target density. This means that SNFs can also perform nonequilibrium moves along the transformation, as done by Langevin dynamics without acceptance step and by the deterministic flow transformations such as RealNVP and neural spline flows.

More closely related is [37] which uses of stochastic flows for density estimation and trains diffusion kernels by maximizing a variational bound of the model likelihood. Their derivation using stochastic paths is similar to ours and this work can be seen as a special instance of SNFs, but it does not consider more general stochastic and deterministic building blocks and does not discuss the problem of asymptotically unbiased sampling of a target density. Ref. [2] proposes a learnable stochastic process by integrating Langevin dynamics with learnable drift and diffusion term. This approach is in a spirit similar as our proposed method, but requires variational approximation of the generative distribution and it has not been worked out how it could be used as a building block within a NF. The approach of [14] combines NF layers with Langevin dynamics, yet approximates the intractable integral with MC samples which we can avoid utilizing the path-weight derivation. Finally, [15] propose a stochastic extension to neural ODEs [3] which can then be trained as samplers. This approach to sampling is very general yet requires costly integration of a SDE which we can avoid by combining simple NFs with stochastic layers.

# 6 Conclusions

We have introduced stochastic normalizing flows (SNFs) that combine both stochastic processes and invertible deterministic transformations into a single learning framework. By leveraging nonequilibrium statistical mechanics we show that SNFs can efficiently be trained to sample asymptotically unbiased from target densities. This can be done by utilizing path probability ratios and avoiding intractabe marginalization. Besides possible applicability in classical machine learning domains such as variational and Bayesian inference, we believe that the latter property can make SNFs a key component in the efficient sampling of many-body physics systems. In future research we aim to apply SNFs with many stochastic sampling steps to accurate large-scale sampling of molecules.

## Broader Impact

The sampling of probability distributions defined by energy models is a key step in the rational design of pharmacological drug molecules for disease treatment, and the design of new materials, e.g., for energy storage. Currently such sampling is mostly done by Molecular Dynamics (MD) and MCMC simulations, which is in many cases limited by computational resources and generates extremely high energy costs. For example, the direct simulation of a single protein-drug binding and dissociation event could require the computational time of an entire supercomputer for a year. Developing machine learning (ML) approaches to solve this problem more efficiently and go beyond existing enhanced sampling methods is therefore of importance for applications in medicine and material science and has potentially far-reaching societal consequences for developing better treatments and reducing energy consumption. Boltzmann Generators, i.e. the combination of Normalizing Flows (FNs) and resampling/reweighting are a new and promising ML approach to this problem and the current paper adds a key technology to overcome some of the previous limitations of NFs for this task.

A risk of the method is that flow-based sampling bears the risk that non-ergodic samplers can be constructed, i.e. samplers that are not guaranteed to sample from the target distribution even in the limit of long simulation time. From such an incomplete sample, wrong conclusions can be drawn. While incomplete sampling is also an issue with MD/MCMC, it is well understood how to at least ensure ergodicity of these methods in the asymptotic limit, i.e. in the limit of generating enough data. Further research is needed to obtain similar results with normalizing flows.

**Acknowledgements.** Special thanks to José Migual Hernández Lobarto (University of Cambridge) for his valuable input on the path probability ratio for MCMC and HMC. We acknowledge funding from the European Commission (ERC CoG 772230 ScaleCell), Deutsche Forschungsgemeinschaft (GRK DAEDALUS, SFB1114/A04), the Berlin Mathematics center MATH+ (Project AA1-6 and EF1-2) and the Fundamental Research Funds for the Central Universities of China (22120200276).

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
