[Supplementary Material]

## Supplementary Material

### 1. Training normalizing flows

**Energy-based training and forward weight maximization.** If the target density $\mu_X$ is known up to a constant $Z_X$, we minimize the forward KL divergence between the generated and the target distribution.

$$
\begin{aligned}
&\mathrm{KL}(p_X \parallel \mu_X) \\
&= \mathbb{E}_{\mathbf{x} \sim p_X(\mathbf{x})} \left[ \log p_X(\mathbf{x}) - \log \mu_X(\mathbf{x}) \right] \\
&= \mathbb{E}_{\mathbf{z} \sim \mu_Z(\mathbf{z})} \left[ u_X(F_{ZX}(\mathbf{z})) - \Delta S_{ZX}(\mathbf{z}) \right] + \mathrm{const.}
\end{aligned}
\tag{23}
$$

The importance weights wrt the target distribution can be computed as:

$$
w_X(\mathbf{x}) = \exp\left( -u_X\left(F_{ZX}(\mathbf{z})\right) + u_Z(\mathbf{z}) + \Delta S_{ZX}(\mathbf{z}) \right) \propto \frac{\mu_X(\mathbf{x})}{p_X(\mathbf{x})}.
\tag{24}
$$

As $\mathbb{E}_{\mathbf{z} \sim p_Z(\mathbf{z})} \left[ u_Z(\mathbf{z}) \right]$ is a constant, we can equivalently minimize KL or maximize log weights:

$$
\max \mathbb{E}_{\mathbf{z} \sim p_Z(\mathbf{z})} \left[ \log w_X(\mathbf{x}) \right] = \min \mathrm{KL}(p_X \parallel \mu_X),
\tag{25}
$$

**Maximum likelihood and backward weight maximization.** The backward KL divergence $\mathrm{KL}(\mu_X \parallel p_X)$ is not always tractable as $\mu_X(\mathbf{x})$ can be difficult to sample from. Replacing $\mu_X(\mathbf{x})$ by the empirical data distribution $\rho_X(\mathbf{x})$, the KL becomes a negative log-likelihood:

$$
\begin{aligned}
&\mathrm{NLL}(\rho_X \parallel p_X) \\
&= \mathbb{E}_{\mathbf{x} \sim \rho_X(\mathbf{x})} \left[ u_Z(F_{XZ}(\mathbf{x})) - \Delta S_{XZ}(\mathbf{x}) \right] + \mathrm{const.}
\end{aligned}
\tag{26}
$$

Using $\mathbb{E}_{\mathbf{x} \sim \rho_X(\mathbf{x})} \left[ -\log \rho_X(\mathbf{x}) \right] = \mathrm{const}$ and the weights:

$$
w_Z(\mathbf{z}) = \exp\left( -u_Z\left(F_{XZ}(\mathbf{x})\right) - \log \rho_X(\mathbf{x}) + \Delta S_{XZ}(\mathbf{x}) \right) \propto \frac{\mu_Z(\mathbf{z})}{p_Z(\mathbf{z})},
$$

maximum likelihood equals log weight maximization:

$$
\max \mathbb{E}_{\mathbf{x} \sim \rho_X(\mathbf{x})} \left[ \log w_Z(\mathbf{z}) \right] = \min \mathrm{NLL}(\rho_X \parallel p_X).
\tag{27}
$$

### 2. Proof of theorem 1 (unbiased sampling with SNF importance weights)

Considering

$$
\begin{aligned}
\mathbb{E}_{\mu_X}[O] &= \int \mu_X(\mathbf{x}) O(\mathbf{x}) \mathrm{d}\mathbf{x} \\
&= \iint \mu_X(\mathbf{x}) \mathbb{P}_b(\mathbf{x} \to \mathbf{z}) O(\mathbf{x}) \mathrm{d}\mathbf{z} \mathrm{d}\mathbf{x} \\
&= \iint \mu_Z(\mathbf{z}) \mathbb{P}_f(\mathbf{z} \to \mathbf{x}) \left[ \frac{\mu_X(\mathbf{x}) \mathbb{P}_b(\mathbf{x} \to \mathbf{z})}{\mu_Z(\mathbf{z}) \mathbb{P}_f(\mathbf{z} \to \mathbf{x})} O(\mathbf{x}) \right] \mathrm{d}\mathbf{z} \mathrm{d}\mathbf{x} \\
&= \mathbb{E}_f \left[ \frac{\mu_X(\mathbf{x}) \mathbb{P}_b(\mathbf{x} \to \mathbf{z})}{\mu_Z(\mathbf{z}) \mathbb{P}_f(\mathbf{z} \to \mathbf{x})} O(\mathbf{x}) \right],
\end{aligned}
$$

where $\mathbb{E}_f$ denotes the expectation over forward path realizations. In practice, we do not know the normalization constant of $\mu_X$ and we therefore replace $\frac{\mu_X(\mathbf{x}) \mathbb{P}_b(\mathbf{x} \to \mathbf{z})}{\mu_Z(\mathbf{z}) \mathbb{P}_f(\mathbf{z} \to \mathbf{x})}$ by the unnormalized path weights in Eq. (11). Then we must normalize the estimator for expectation values, obtaining:

$$
\frac{\sum_{k=1}^N \mathbf{w}(\mathbf{z}_k \to \mathbf{x}_k) O(\mathbf{x}_k)}{\sum_{k=1}^N \mathbf{w}(\mathbf{z}_k \to \mathbf{x}_k)} \xrightarrow{p} \mathbb{E}_\mu[O]
$$

which converges towards $\mathbb{E}_\mu[O]$ with $N \to \infty$ according to the law of large numbers.

## 3. Derivation of the deterministic layer probability ratio

In order to work with delta distributions, we define $\delta^\sigma(\mathbf{x}) = \mathcal{N}(\mathbf{x}; \mathbf{0}, \sigma\mathbf{I})$, i.e. a Gaussian normal distribution with mean $\mathbf{0}$ and variance $\sigma$ and then consider the limit $\sigma \to 0^+$. In the case where $\sigma > 0$, by defining

$$q_t^\sigma(\mathbf{y}_t \to \mathbf{y}_{t+1}) = \delta^\sigma(\mathbf{y}_{t+1} - F_t(\mathbf{y}_t)),$$

and

$$
\begin{aligned}
\tilde{q}_t^\sigma(\mathbf{y}_{t+1} \to \mathbf{y}_t) &= \frac{p_t(\mathbf{y}_t)q_t^\sigma(\mathbf{y}_t \to \mathbf{y}_{t+1})}{\int p_t(\mathbf{y})q_t^\sigma(\mathbf{y} \to \mathbf{y}_{t+1})\mathrm{d}\mathbf{y}} \\
&= \frac{p_t(\mathbf{y}_t)\delta^\sigma(\mathbf{y}_{t+1} - F_t(\mathbf{y}_t))}{\int p_t(\mathbf{y})\delta^\sigma(\mathbf{y}_{t+1} - F_t(\mathbf{y}))\mathrm{d}\mathbf{y}},
\end{aligned}
$$

we have

$$\frac{\tilde{q}_t^\sigma(\mathbf{y}_{t+1} \to \mathbf{y}_t)}{q_t^\sigma(\mathbf{y}_t \to \mathbf{y}_{t+1})} = \frac{p_t(\mathbf{y}_t)}{\int p_t(\mathbf{y})\delta^\sigma(\mathbf{y}_{t+1} - F(\mathbf{y}))\mathrm{d}\mathbf{y}},$$

where $p_t(y_t)$ denotes the marginal distribution of $\mathbf{y}_t$. By considering

$$
\begin{aligned}
\lim_{\sigma \to 0^+} \int p_t(\mathbf{y})\delta^\sigma(\mathbf{y}_{t+1} - F_t(\mathbf{y}))\mathrm{d}\mathbf{y} &= \lim_{\sigma \to 0^+} \int p_t(F_t^{-1}(\mathbf{y}'))\delta^\sigma(\mathbf{y}_{t+1} - \mathbf{y}') \left| \det\left( \frac{\partial F_t^{-1}(\mathbf{y}')}{\partial \mathbf{y}'} \right) \right| \mathrm{d}\mathbf{y}' \\
&= p_t(F_t^{-1}(\mathbf{y}_{t+1})) \left| \det\left( \frac{\partial F_t^{-1}(\mathbf{y}_{t+1})}{\partial \mathbf{y}_{t+1}} \right) \right| \\
&= p_t(\mathbf{y}_t) \left| \det \mathbf{J}_t(\mathbf{y}_t) \right|^{-1}
\end{aligned}
$$

and using the definition of $\Delta S_t$ in terms of path probability rations, we obtain:

$$
\begin{aligned}
\exp\left( \Delta S_t \right) = \frac{\tilde{q}_t(\mathbf{y}_{t+1} \to \mathbf{y}_t)}{q_t(\mathbf{y}_t \to \mathbf{y}_{t+1})} &= \lim_{\sigma \to 0^+} \frac{\tilde{q}_t^\sigma(\mathbf{y}_{t+1} \to \mathbf{y}_t)}{q_t^\sigma(\mathbf{y}_t \to \mathbf{y}_{t+1})} \\
&= \lim_{\sigma \to 0^+} \frac{p_t(\mathbf{y}_t)}{\int p_t(\mathbf{y})\delta^\sigma(\mathbf{y}_{t+1} - F(\mathbf{y}))\mathrm{d}\mathbf{y}} \\
&= \left| \det \mathbf{J}_t(\mathbf{y}_t) \right|
\end{aligned}
$$

and thus

$$\Delta S_t = \log\left| \det \mathbf{J}_t(\mathbf{y}_t) \right|.$$

## 4. Derivation of the overdamped Langevin path probability ratio

These results follow [31]. The backward step is realized by

$$\mathbf{y}_t = \mathbf{y}_{t+1} - \epsilon_t \nabla u_\lambda(\mathbf{y}_{t+1}) + \sqrt{\frac{2\epsilon}{\beta}} \tilde{\boldsymbol{\eta}}_t. \tag{28}$$

Combining Equations (18) and (28):

$$-\epsilon_t \nabla u_\lambda(\mathbf{y}_t) + \sqrt{\frac{2\epsilon_t}{\beta}} \boldsymbol{\eta}_t = \epsilon_t \nabla u_\lambda(\mathbf{y}_{t+1}) - \sqrt{\frac{2\epsilon_t}{\beta}} \tilde{\boldsymbol{\eta}}_t.$$

and thus

$$\tilde{\boldsymbol{\eta}}_t = \sqrt{\frac{\epsilon_t \beta}{2}} \left[ \nabla u_\lambda(\mathbf{y}_t) + \nabla u_\lambda(\mathbf{y}_{t+1}) \right] - \boldsymbol{\eta}_t.$$

Resulting in the path probability ratio:

$$
\begin{aligned}
\exp\left( \Delta S_t \right) = \frac{q_t(\mathbf{y}_{t+1} \to \mathbf{y}_t)}{q_t(\mathbf{y}_t \to \mathbf{y}_{t+1})} &= \frac{p(\tilde{\boldsymbol{\eta}}_t) \left| \frac{\partial \mathbf{y}_t}{\partial \tilde{\boldsymbol{\eta}}_t} \right|}{p(\boldsymbol{\eta}_t) \left| \det\left( \frac{\partial \mathbf{y}_{t+1}}{\partial \boldsymbol{\eta}_t} \right) \right|} \\
&= \frac{p(\tilde{\boldsymbol{\eta}}_t)}{p(\boldsymbol{\eta}_t)} = e^{-\frac{1}{2}\left( \|\tilde{\boldsymbol{\eta}}_t\|^2 - \|\boldsymbol{\eta}_t\|^2 \right)}.
\end{aligned}
$$

and thus

$$-\Delta S_t = \frac{1}{2}\left( \|\tilde{\boldsymbol{\eta}}_t\|^2 - \|\boldsymbol{\eta}_t\|^2 \right)$$

## 5. Derivation of the Langevin probability ratio

These results follow [31]. We define constants:

$$c_1 = \frac{\Delta t}{2m}$$

$$c_2 = \sqrt{\frac{4\gamma m}{\Delta t \beta}}$$

$$c_3 = 1 + \frac{\gamma \Delta t}{2}$$

Then, the forward step of Brooks-Brünger-Karplus (BBK, leap-frog) Langevin dynamics are defined as:

$$\mathbf{v}' = \mathbf{v}_t + c_1 \left[ -\nabla u_\lambda(\mathbf{x}_t) - \gamma m \mathbf{v}_t + c_2 \boldsymbol{\eta}_t \right] \tag{29}$$

$$\mathbf{x}_{t+1} = \mathbf{x}_t + \Delta t \mathbf{v}' \tag{30}$$

$$\mathbf{v}_{t+1} = \frac{1}{c_3} \left[ \mathbf{v}' + c_1 \left( -\nabla u_\lambda(\mathbf{x}_{t+1}) + c_2 \boldsymbol{\eta}'_t \right) \right] \tag{31}$$

Note that the factor 4 in sqrt is different from [31] – this factor is needed as we employ $\Delta t/2$ in both half-steps. The backward step with reversed momenta, $(\mathbf{x}_{t+1}, -\mathbf{v}_{t+1}) \to (\mathbf{x}_t, -\mathbf{v}_t)$ is then defined by:

$$\mathbf{v}'' = -\mathbf{v}_{t+1} + c_1 \left[ -\nabla u_\lambda(\mathbf{x}_{t+1}) + \gamma m \mathbf{v}_{t+1} + c_2 \tilde{\boldsymbol{\eta}}_t \right] \tag{32}$$

$$\mathbf{x}_t = \mathbf{x}_{t+1} + \Delta t \mathbf{v}'' \tag{33}$$

$$-\mathbf{v}_t = \frac{1}{c_3} \left[ \mathbf{v}'' + c_1 \left( -\nabla u_\lambda(\mathbf{x}_t) + c_2 \tilde{\boldsymbol{\eta}}'_t \right) \right] \tag{34}$$

To compute the momenta $\tilde{\boldsymbol{\eta}}_t, \tilde{\boldsymbol{\eta}}'_t$ that realize the reverse step, we first combine Eqs. (30-33) to obtain:

$$\mathbf{v}' = -\mathbf{v}'' \tag{35}$$

Combining Eqs. (31), (32) and (35), we obtain:

$$\left( 1 + \frac{\gamma \Delta t}{2} \right) \mathbf{v}_{t+1} = \mathbf{v}' + c_1 \left( -\nabla u_\lambda(\mathbf{x}_{t+1}) + c_2 \boldsymbol{\eta}'_t \right)$$

$$\left( 1 - \frac{\gamma \Delta t}{2} \right) \mathbf{v}_{t+1} = \mathbf{v}' + c_1 \left( -\nabla u_\lambda(\mathbf{x}_{t+1}) + c_2 \tilde{\boldsymbol{\eta}}_t \right),$$

and:

$$\tilde{\boldsymbol{\eta}}_t = \boldsymbol{\eta}'_t - \sqrt{\gamma \Delta t m \beta} \mathbf{v}_{t+1}$$

Combining Eqs. (29), (34) and (35), we obtain:

$$-\mathbf{v}_t \left( 1 - \frac{\gamma \Delta t}{2} \right) = \mathbf{v}'' + c_1 \left( -\nabla u_\lambda(\mathbf{x}_t) + c_2 \boldsymbol{\eta}_t \right)$$

$$-\mathbf{v}_t \left( 1 + \frac{\gamma \Delta t}{2} \right) = \mathbf{v}'' + c_1 \left( -\nabla u_\lambda(\mathbf{x}_t) + c_2 \tilde{\boldsymbol{\eta}}'_t \right),$$

and:

$$-\mathbf{v}_t \left( 1 - \frac{\gamma \Delta t}{2} \right) - c_2 \boldsymbol{\eta}_t = -\mathbf{v}_t \left( 1 + \frac{\gamma \Delta t}{2} \right) - c_2 \tilde{\boldsymbol{\eta}}'_t$$

$$\tilde{\boldsymbol{\eta}}'_t = \boldsymbol{\eta}_t - \sqrt{\gamma \Delta t m \beta} \mathbf{v}_t$$

To compute the path probability ratio we introduce the Jacobian

$$J(\boldsymbol{\eta}_t, \boldsymbol{\eta}'_t) = \det \begin{bmatrix} \frac{\partial \mathbf{x}_{t+1}}{\partial \boldsymbol{\eta}_t} & \frac{\partial \mathbf{v}_{t+1}}{\partial \boldsymbol{\eta}_t} \\ \frac{\partial \mathbf{x}_{t+1}}{\partial \boldsymbol{\eta}'_t} & \frac{\partial \mathbf{v}_{t+1}}{\partial \boldsymbol{\eta}'_t} \end{bmatrix}$$

and find:

$$\exp\left(\Delta S_t\right) = \frac{\tilde{q}_t\left((\mathbf{x}_{t+1}, -\mathbf{v}_{t+1}) \to (\mathbf{x}_t, \mathbf{v}_t)\right)}{q_t\left((\mathbf{x}_t, \mathbf{v}_t) \to (\mathbf{x}_{t+1}, -\mathbf{v}_{t+1})\right)}$$

$$= \frac{p(\tilde{\boldsymbol{\eta}}_t)p(\tilde{\boldsymbol{\eta}}_t')J(\tilde{\boldsymbol{\eta}}_t, \tilde{\boldsymbol{\eta}}_t')}{p(\boldsymbol{\eta}_t)p(\boldsymbol{\eta}_t')J(\boldsymbol{\eta}_t, \boldsymbol{\eta}_t')}$$

$$-\Delta S_t = \frac{1}{2}\left(\left(\|\tilde{\boldsymbol{\eta}}_t\|^2 + \|\tilde{\boldsymbol{\eta}}_t'\|^2\right) - \left(\|\boldsymbol{\eta}_t\|^2 + \|\boldsymbol{\eta}_t'\|^2\right)\right)$$

where the Jacobian ratio cancels as the Jacobians are independent of the noise variables.

## 6. Derivation of the probability ratio for Markov Chain Monte Carlo

For MCMC, $q_t$ satisfies the detailed balance condition

$$\exp(-u_\lambda(\mathbf{y}_t)) \cdot q_t(\mathbf{y}_t \to \mathbf{y}_{t+1}) = \exp(-u_\lambda(\mathbf{y}_{t+1})) \cdot \tilde{q}_t(\mathbf{y}_{t+1} \to \mathbf{y}_t)$$

with respect to the potential function $u_\lambda$. We have

$$
\begin{aligned}
\Delta S_t &= \log \frac{\tilde{q}_t(\mathbf{y}_{t+1} \to \mathbf{y}_t)}{q_t(\mathbf{y}_t \to \mathbf{y}_{t+1})} \\
&= \log \frac{\exp(-u_\lambda(\mathbf{y}_t))}{\exp(-u_\lambda(\mathbf{y}_{t+1}))} \\
&= u_\lambda(\mathbf{y}_{t+1}) - u_\lambda(\mathbf{y}_t)
\end{aligned}
$$

## 7. Derivation of the probability ratio for Hamiltonian MC with Metropolis acceptance

Hamiltonian MC with Metropolis acceptance defines a forward path density

$$q_t\left((\mathbf{y}_t, \mathbf{v}) \to (\mathbf{y}_{t+1}, \mathbf{v}^K)\right)$$

which satisfies the joint detailed balance condition

$$\exp(-u_\lambda(\mathbf{y}_t))\mathcal{N}(\mathbf{v}|\mathbf{0}, \mathbf{I}) \cdot q_t\left((\mathbf{y}_t, \mathbf{v}) \to (\mathbf{y}_{t+1}, \mathbf{v}^K)\right)$$
$$= \exp(-u_\lambda(\mathbf{y}_{t+1}))\mathcal{N}(\mathbf{v}^K|\mathbf{0}, \mathbf{I}) \cdot \tilde{q}_t\left((\mathbf{y}_{t+1}, \mathbf{v}^K) \to (\mathbf{y}_t, \mathbf{v})\right). \qquad (36)$$

Considering the velocity $\mathbf{v}$ is independently drawn from $\mathcal{N}(\mathbf{v}|\mathbf{0}, \mathbf{I})$, the "marginal" forward path density of $\mathbf{y}_t \to \mathbf{y}_{t+1}$ is

$$q_t\left(\mathbf{y}_t \to \mathbf{y}_{t+1}\right) = \iint \mathcal{N}(\mathbf{v}|\mathbf{0}, \mathbf{I}) \cdot q_t\left((\mathbf{y}_t, \mathbf{v}) \to (\mathbf{y}_{t+1}, \mathbf{v}^K)\right) \mathrm{d}\mathbf{v}\mathrm{d}\mathbf{v}^K.$$

Then, it can be obtained from (36) that

$$
\begin{aligned}
\exp(-u_\lambda(\mathbf{y}_t))q_t\left(\mathbf{y}_t \to \mathbf{y}_{t+1}\right) &= \iint \exp(-u_\lambda(\mathbf{y}_t))\mathcal{N}(\mathbf{v}|\mathbf{0}, \mathbf{I})\left((\mathbf{y}_t, \mathbf{v}) \to (\mathbf{y}_{t+1}, \mathbf{v}^K)\right) \mathrm{d}\mathbf{v}\mathrm{d}\mathbf{v}^K \\
&= \iint \exp(-u_\lambda(\mathbf{y}_{t+1}))\mathcal{N}(\mathbf{v}^K|\mathbf{0}, \mathbf{I})\tilde{q}_t\left((\mathbf{y}_{t+1}, \mathbf{v}^K) \to (\mathbf{y}_t, \mathbf{v})\right) \mathrm{d}\mathbf{v}\mathrm{d}\mathbf{v}^K \\
&= \exp(-u_\lambda(\mathbf{y}_{t+1}))\tilde{q}_t\left(\mathbf{y}_{t+1} \to \mathbf{y}_t\right),
\end{aligned}
$$

and

$$
\begin{aligned}
\Delta S_t &= \log \frac{\tilde{q}_t\left(\mathbf{y}_{t+1} \to \mathbf{y}_t\right)}{q_t\left(\mathbf{y}_t \to \mathbf{y}_{t+1}\right)} \\
&= u_\lambda(\mathbf{y}_{t+1}) - u_\lambda(\mathbf{y}_t)
\end{aligned}
$$

## 8. Details on using SNFs for variational inference

Here we elaborate on the details of using SNFs as a variational approximation of the posterior distribution of a *variational autoencoder* (VAE) [21] as presented in our last results section. In contrast to the usual notation used in common VAE literature, we choose $\mathbf{x}$ to indicate the *latent*

variable, while we call the *observed* variable $\mathbf{s}$. This is due to being consistent with the use of $\mathbf{x}$ as the sampled variable of interest throughout our former discussions.

For a given data set $\{\mathbf{s}_1, \ldots, \mathbf{s}_N\}$, the decoder $D$ of a VAE characterizes each $\mathbf{s}$ as a random variable with a tractable distribution $\mathbb{P}_D(\mathbf{s}|\mathbf{x})$ depending on a unknown latent variable $\mathbf{x}$. Furthermore, the prior distribution is assumed to be tractable as well (e.g. an isotropic normal distribution). Here we take the prior

$$\mathbb{P}(\mathbf{x}) = \mathcal{N}(\mathbf{x} \mid \mathbf{0}, \mathbf{I}).$$

Together, this defines the joint distribution

$$\mathbb{P}_D(\mathbf{x}, \mathbf{s}) = \mathbb{P}(\mathbf{x}) \cdot \mathbb{P}_D(\mathbf{s}|\mathbf{x}).$$

Conditioned on a given $\mathbf{s}$, we can utilize a SNF to approximate the posterior distribution

$$\mathbb{P}_D(\mathbf{x}|\mathbf{s}) = \frac{\mathbb{P}_D(\mathbf{x}, \mathbf{s})}{\mathbb{P}_D(\mathbf{s})}.$$

For convenience and consistency with the former discussion, we define $\mu_X(\mathbf{x}) = \mathbb{P}_D(\mathbf{x}|\mathbf{s})$ and $u_X(\mathbf{x}) = -\log \mathbb{P}_D(\mathbf{x}, \mathbf{s})$. Thus, the parameters of the SNF and the decoder $D$ can be trained by minimizing $J_{KL}$ which provides an upper bound of the negative log-likelihood of $\mathbf{s}$ as follows:

$$
\begin{aligned}
J_{KL}(\mathbf{s}) &= \mathbb{E}_{\mathbf{z} \sim \mu_Z, \mathbf{y}_1, \ldots, \mathbf{y}_T}[u_X(\mathbf{y}_T) - \sum_{t=0}^{T-1} \Delta S_t] \\
&= \mathbb{E}_{\mathbf{z} \sim \mu_Z, \mathbf{y}_1, \ldots, \mathbf{y}_T}[-\log \mu_X(\mathbf{y}_T) - \sum_{t=0}^{T-1} \Delta S_t] - \log \mathbb{P}_D(\mathbf{s}) \\
&= \mathrm{KL}(\mu_Z(\mathbf{z})\mathbb{P}_f(\mathbf{z} \to \mathbf{x}) \parallel \mu_X(\mathbf{x})\mathbb{P}_b(\mathbf{x} \to \mathbf{z})) - \log \mathbb{P}_D(\mathbf{s}) \\
&\geq -\log \mathbb{P}_D(\mathbf{s})
\end{aligned}
$$

If the SNF consists of only deterministic transformations, $J_{KL}$ is equivalent to $\mathcal{F}$ in [35].

We estimate $J_{KL}(\mathbf{s})$ on samples as

$$\hat{J}_{KL}(\mathbf{s}) = \frac{1}{M} \sum_{i=1}^{M} u_X(\mathbf{y}_T^{(i)}) - \sum_{t=0}^{T-1} \Delta S_t^{(i)}, \tag{37}$$

by sampling $M$ paths $\{(\mathbf{y}_0^{(i)}, \ldots, \mathbf{y}_T^{(i)})\}_{i=1}^{M}$ for each $\mathbf{s}$ and setting $M = 5$.

**Estimating the evidence.** After training we approximate $-\log \mathbb{P}_D(\mathbf{s})$ by marginalizing out the latent variable $\mathbf{x}$ via Monte Carlo sampling. In order to improve sampling efficiency and have a fair comparison among the three different SNF instantiations, we approximate the posterior distribution $\mathbb{P}_D(\mathbf{x}|\mathbf{s})$ of the trained model using the same variational approximation:

1. We define a simple base distribution $q(\mathbf{z}) = \mathcal{N}(\mathbf{z}|\mathbf{0}, \mathbf{I})$, together with a conditional diffeomorphism $F_{LL}(\mathbf{x}|\mathbf{s})$ transforming $\mathbf{z}$ to $\mathbf{x}$ and vice-versa conditioned on $\mathbf{s}$:

$$\mathbf{x} \quad \underset{F_{LL}(\cdot|\mathbf{s})}{\overset{F_{LL}^{-1}(\cdot|\mathbf{s})}{\rightleftarrows}} \quad \mathbf{z}.$$

   We realize such a conditional flow via RealNVP transformations, where coupling layers are additionally conditioned on $\mathbf{s}$ and only $\mathbf{x}/\mathbf{z}$ is transformed during the flow. Together with $q(\mathbf{z})$ this defines the conditional distribution

$$q_{LL}(\mathbf{x}|\mathbf{s}) = q(F_{LL}^{-1}(\mathbf{x}|\mathbf{s})) \left| \det \left( \frac{\partial F_{LL}^{-1}(\mathbf{x}|\mathbf{s})}{\partial \mathbf{x}} \right) \right|$$

   which we use as variational approximation to the true posterior. We then train $q_{LL}$ by minimizing the KL divergence

$$\mathbb{E}_{\mathbf{s}, \mathbf{z} \sim q(\mathbf{z})} \left[\log q_{LL}(F_{LL}(\mathbf{z}|\mathbf{s})|\mathbf{s}) - \log \mathbb{P}(F_{LL}(\mathbf{z}|\mathbf{s})) - \log \mathbb{P}_D(\mathbf{s}|F_{LL}(\mathbf{z}|\mathbf{s}))\right] + const.$$

   until convergence. This loss is minimized iff $q_{LL}(\mathbf{x}|\mathbf{s}) = \mathbb{P}_D(\mathbf{x}|\mathbf{s})$.

2. Now considering

$$\mathbb{P}_D(\mathbf{s}) = \int \mathbb{P}(\mathbf{x})\mathbb{P}_D(\mathbf{s}|\mathbf{x})\mathrm{d}\mathbf{x}$$

$$= \int q_{LL}(\mathbf{x}|\mathbf{s})\frac{\mathbb{P}(\mathbf{x})\mathbb{P}_D(\mathbf{s}|\mathbf{x})}{q_{LL}(\mathbf{x}|\mathbf{s})}\mathrm{d}\mathbf{x}$$

$$= \mathbb{E}_{\mathbf{x}\sim q_{LL}(\mathbf{x}|\mathbf{s})}\left[\frac{\mathbb{P}(\mathbf{x})\mathbb{P}_D(\mathbf{s}|\mathbf{x})}{q_{LL}(\mathbf{x}|\mathbf{s})}\right]$$

$$= \mathbb{E}_{\mathbf{z}\sim\mathcal{N}(\mathbf{z}|\mathbf{0},\mathbf{I})}\left[\frac{\mathbb{P}(F_{LL}(\mathbf{z}|\mathbf{s}))\mathbb{P}_D(\mathbf{s}|F_{LL}(\mathbf{z}|\mathbf{s}))}{q(\mathbf{z})\left|\det\left(\frac{\partial F_{LL}^{-1}(\mathbf{x}|\mathbf{s})}{\partial\mathbf{x}}\right)\right|^{-1}}\right],$$

we can draw $N$ samples $\mathbf{z}^{(1)},\ldots,\mathbf{z}^{(N)}$ and approximate $\mathbb{P}_D(\mathbf{s})$ by

$$\hat{\mathbb{P}}_D(\mathbf{s}) = \frac{1}{N}\sum_{i=1}^{N}\frac{\mathbb{P}(F_{LL}(\mathbf{z}^{(\mathbf{i})}|\mathbf{s}))\mathbb{P}_D(\mathbf{s}|F_{LL}(\mathbf{z}^{(i)}|\mathbf{s}))}{q(\mathbf{z}^{(i)})\left|\det\left(\frac{\partial F_{LL}^{-1}(\mathbf{x}|\mathbf{s})}{\partial\mathbf{x}}\right)\right|^{-1}}.$$

In experiments, $N$ is set to be 2000.

In table 3, the first column is the mean value of $\hat{J}_{KL}(\mathbf{s})$ on the test data set as a variational bound of the mean of $-\log p(\mathbf{s})$ (related to Fig. 4a in [35]). The second column is the mean value of $-\log\hat{\mathbb{P}}_D(\mathbf{s})$ on the test data set (related to Fig. 4c in [35]).

## 9. Hyper-parameters and other benchmark details

All experiments were run using PyTorch 1.2 and on GTX1080Ti cards. Optimization uses Adam [20] with step-size 0.001 and otherwise default parameters. All deterministic flow transformations use RealNVP [6]. A RealNVP block is defined by two subsequent RealNVP layers that are swapped such that each channel gets transformed once as a function of the other channel. The affine transformation of each RealNVP layer is given by a fully connected ReLU network. For the NSF layers we substitute the simple affine transformations used in RealNVP by the rational-quadratic (RQ) spline transformation implemented in `https://github.com/bayesiains/nflows`. As before the width, height and slope of the RQ transformations are given by fully connected ReLU networks. Again a NSF block consists of two subsequent NSF layers with intermediate swap layers.

**Double well examples in Figures 1 and 4**

- Both normalizing flow and SNF networks use 3 RealNVP blocks with three hidden layers of dimension 64. The SNF additionally uses 20 Metropolis MC steps per block using a Gaussian proposal density with standard deviation 0.25.
- Training is done by minimizing $J_{ML}$ for 300 iterations and $\frac{1}{2}J_{ML} + \frac{1}{2}KL$ for 300 iterations using a batch-size of 128.
- "Biased data" is defined by running local Metropolis MC in each of the two wells. These simulations do not transition to the other well and we use 1000 data points in each well for training.
- "Unbiased data" is produced by running Metropolis MC with a large proposal step (standard deviation 1.5) to convergence and retaining 10000 data points for training.
- In Table S1, the sampling results of SNFs with RealNVP blocks and Metropolis MC steps. MC step sizes of the first SNF is fixed to be 0.25 as before, and all step sizes of the second one are trainable parameters in $[0.01, 0.3]$. The other settings are the same as in Table 1.

**Two-dimensional image densities in Figure 3**

- RealNVP and NSF flows both use 5 blocks. All involved transformation parameters (transla-tion/scale in RealNVP layers, width/height/slope in NSF layers) use three hidden layers of dimension 64. For the NSF layers we used 20 knot points in the RQ-spline transformation. Training was done by minimizing $J_{ML}$ for 2000 iterations with batch-size 250.

Table S1: Unbiased sampling for double well potential by SNFs with nontrainable/trainable MC step sizes.

| | not reweighted | | | reweighted | | |
|---|---|---|---|---|---|---|
| | bias | $\sqrt{\text{var}}$ | $\sqrt{\text{bias}^2+\text{var}}$ | bias | $\sqrt{\text{var}}$ | $\sqrt{\text{bias}^2+\text{var}}$ |
| RNVP + MC | $1.5 \pm 0.2$ | $0.3 \pm 0.1$ | $1.5 \pm 0.2$ | $0.2 \pm 0.1$ | $0.6 \pm 0.1$ | $0.6 \pm 0.1$ |
| **RNVP + MC with trainable step sizes** | $1.0 \pm 0.2$ | $0.2 \pm 0.1$ | $1.0 \pm 0.2$ | $0.1 \pm 0.1$ | $0.4 \pm 0.1$ | $0.4 \pm 0.1$ |

- Purely stochastic flow (column 2) uses five blocks with 10 Metropolis MC steps each using a Gaussian proposal density with standard deviation 0.1.
- SNF (column 3/5) uses 5 blocks (RNVP/NSF block and 10 Metropolis MC steps with same parameters as above). Training was done by minimizing $J_{ML}$ for 6000 iterations with batch-size 250.

**Alanine dipeptide in Fig. 5**

- Normalizing flow uses 3 RealNVP blocks with 3 hidden layers and $[128, 128, 128]$ nodes in their transformers. Training was done by minimizing $J_{ML}$ for 1000 iterations with batch-size 256.
- SNF uses the same architecture and training parameters, but additionally 20 Metropolis MC steps each using a Gaussian proposal density with standard deviation 0.1.
- As a last flow layer before $\mathbf{x}$, we used an invertible transformation between Cartesian coordinates and internal coordinates (bond lengths, angles, torsion angles) following the procedure described in [32]. The internal coordinates were normalized by removing the mean and dividing by the standard deviation of their values in the training data.
- Training data: We set up Alanine dipeptide in vacuum using OpenMMTools. Parameters are defined by the force field ff96 of the AMBER program [34]. Simulations are run at standard OpenMMTools parameters with no bond constraints, 1 femtosecond time-step for $10^6$ time-steps (1 nanosecond) at a temperature of 1000 K in order to facilitate rapid exploration of the $\phi/\psi$ torsion angles and a few hundred transitions between metastable states. $10^5$ atom positions were saved as training data.

**MNIST and Fashion-MNIST VAE in Table 3**

- The latent space dimension was set to 50. The decoder consists of 2 fully connected hidden layers, with 1024 units and ReLU non-linearities for each hidden layer. The activation function of the the output layer is sigmoid function. $\mathbb{P}_D(\mathbf{s}|\mathbf{x})$ is defined as

$$\log \mathbb{P}_D(\mathbf{s}|\mathbf{x}) = \log \mathbb{P}_D(\mathbf{s}|D(\mathbf{x}))$$
$$= \sum_{i=1}^{784} [\mathbf{s}]_i \log[D(\mathbf{x})]_i + (1 - [\mathbf{s}]_i) \log(1 - [D(\mathbf{x})]_i),$$

where $[\mathbf{s}]_i$, $[D(\mathbf{x})]_i$ denote the $i$th pixel of $\mathbf{s}$ and the $i$th output of $D$.

- Adam algorithm is used to train all models. Training was done by minimizing $\hat{J}_{KL}$ (see (37)) for 40 epochs with batch-size 128 and step size $10^{-3}$ unless otherwise stated.
- In simple VAE, the encoder $E$ consists of 2 fully connected hidden layers, with 1024 nodes and ReLU non-linearities for each hidden layer. The encoder has 100 outputs, where the activation function of the first 50 outputs is the linear function and the activation function of the last 50 outputs is the absolute value function. The transformation from $\mathbf{z}$ to $\mathbf{x}$ is given by

$$[\mathbf{x}]_i = [E(\mathbf{s})]_i + [\mathbf{z}]_i \cdot [E(\mathbf{s})]_{i+50}. \tag{38}$$

- MCMC uses 30 Metropolis MC steps each using a overdamped Langevin proposal, where the interpolated potential are used. The interpolation coefficients and the step size of the proposal are both trained as parameters of the flow.

- Normalizing flow uses 6 RealNVP blocks with 2 hidden layers and $[64, 64]$ nodes in their transformers.
- SNF uses three units with each unit consisting of 2 RealNVP blocks + 10 Metropolis MC steps, where architectures are the same as the above. During the training procedure, we first train parameters of the 6 RealNVP blocks without the Metropolis MC steps for 20 epochs, and then train all parameters for another 20 epochs. The training step size is $10^{-3}$ for the first 20 epochs and $10^{-4}$ for the last 20 epochs.
- For calculating the marginal likelihood $\mathbb{P}_D(\mathbf{s})$, $F_{LL}$ uses 12 RealNVP blocks with 2 hidden layers and $[256, 256]$ nodes in their transformers.

## 10. Comparison with related sampling methods

A brief comparison of the proposed SNF and selected sampling methods with learnable proposals and transformations is provided in Table S2. Most previous sampling methods are developed based on the detailed balance in each step, except that HVI presented in [36] can perform nonequilibrium sampling steps by using annealed target distributions. Furthermore, some sampling techniques [16] and [17] also improve the sampling efficiency by linear or nonlinear deterministic transformation, where the transformation is performed only once. It can be seen from the comparison that SNF provides a universal framework for sampling, where the deterministic and stochastic blocks can be flexibly designed and combined.

Table S2: A comparison of samplers with learnable proposals/transformations.

| Method | Containing nonequilibrium sampling steps | Combining with learnable deterministic transformation |
|---|---|---|
| HVI [36] | ✓ | ✗ |
| L2HMC [24] | ✗ | ✗ |
| A-NICE-MC [38] | ✗ | ✗ |
| HMC for DLGMs [16] | ✗ | ✓ |
| NeuTra [17] | ✗ | ✓ |
| SNF | ✓ | ✓ |

**Supplementary Figures**

Figure S1: **Reproducibility of normalizing flows for the double well**. Red arrows indicate deterministic transformations (perturbations), blue arrows indicate stochastic dynamics (relaxations). **a-b**) Two independent runs of 3 RealNVP blocks (6 layers). **c-d**) Two independent runs of same architecture with 20 BD steps before/after RealNVP blocks.

Figure S2: **Computational cost of adding stochastic layers**. Time required for training SNFs of images shown in 3 with a fixed number of steps, as a function of the number of stochastic layers per RNVP or NSF layer. Timings are normalized to one RNVP or NSF layer. While details of these timings depend on hyperparameters, implementation and compute platform, the main feature is that deterministic flow layers are much more computationally expensive than stochastic flow layers, and therefore a few stochastic flow layers can be added to each deterministic flow layer without significant increase in computational cost.