[Reviews · NeurIPS 2020]

Review 1

Summary and Contributions: The paper presents a new class of generative models that combine normalizing flows with stochastic dynamics. Borrowing ideas from nonequilibrium statistical physics, the authors develop an efficient training procedure that does not require explicit marginalization over stochastic variables. Instead, a KL is minimized in the path-space of each flow trajectory. The method they present is a neural extension of Nonequilibrium Candidate Monte Carlo, a computational scheme developed in the field of nonequilibrium statistical physics. Their numerical experiments demonstrate increased expressivity and efficiency. While flow models struggle to model multi-modal distributions due to topological constraints, their results show that stochastic dynamics can aid in resolving these modes. They further demonstrate gains in sample efficiency over pure MCMC and RealNVP baselines.

Strengths: This paper presents a novel contribution to generative models: A model admitting both flow- and stochastic-blocks, and a tractable training procedure that does not require marginalization of stochastic blocks. It builds a bridge between ML and computational nonequilibrium statistical physics. These connections can be very valuable for both communities.

Weaknesses: The manuscript exposition is heavily reliant on literature from nonequilibrium statistical physics. This could be hard to digest for some. The authors showed that their method achieves higher sample efficiency compared to baselines. But the method also requires more compute (due to the stochastic blocks). I would be interested in seeing an efficiency comparison along this axis. The baselines are pretty simple. I would’ve liked to see comparisons with e.g. Neural Spline Flows, which have shown to outperform older models like RealNVP.

Correctness: All claims, methods, methodology seem correct.

Clarity: My main concern with the manuscript exposition is the heavy reliance on previous literature from nonequilibrium statistical physics. I wonder how digestible this would be for a general ML audience without the background knowledge.

Relation to Prior Work: Yes

Reproducibility: Yes

Additional Feedback: Nice work! We need more bridge-building between physics and ML. This is a nice example. In the Eq. above line 39, $rho_X$ is not defined.


Review 2

Summary and Contributions: This paper presents a framework in which deterministic normalising flows can be expanded to include stochastic mappings. The paper outlines an associated forward-backward ratio analogous to the deterministic change of variables, which sidesteps the need to marginalize over all stochastic paths, and includes the deterministic change of variables as a limiting case. The paper then examines the use of Langevin and MCMC stochastic dynamics as building blocks in this framework, and explores their effectiveness in learning known target densities in illustrative toy tasks as well as higher-dimensional applications in molecular dynamics.

Strengths: The theoretical claims seem correct, and the empirical evaluation demonstrates the usefulness of the method over (somewhat) reasonable baselines. Couching deterministic flows as a limiting case of stochastic transitions, as well as providing an analogous 'change-of-variables' as a forward-backward ratio, is a strong contribution, and will likely lead to interesting work at the confluence of traditional deterministic flows, diffusion models, and latent variable models in the VAE framework.

Weaknesses: Obviously it's not possible to compare to all flows, and Real NVP provides a reasonable enough starting point, but I have slight concern over its strength as a baseline. Various advances have been made in improving the flexibility of coupling layers, as well as addressing topological considerations with mixture methods, and it would have been nice to really push the strength of the baseline using these. The stochastic framework introduced in the paper is worthwhile in its own right, but the experimental validation is maybe slightly lacking in this respect. Otherwise, no major weaknesses, just some points in the additional feedback.

Correctness: The theoretical claims seem correct, with detailed derivations given in the appendix. The empirical results seem correct, and demonstrate the validity and benefit of the theoretical framework. 137: 'Representational power versus sampling efficiency': I particularly appreciated this section, using a toy example to clearly demonstrate the limitations of existing methods, as well as how these limitations are successfully addressed by the proposed method.

Clarity: The paper is generally well-written, and carries a clear narrative throughout. I've mentioned some points in the additional feedback. 66-70: This nicely encapsulates the central purpose of the paper.

Relation to Prior Work: Discussion of related work is good, and linking to neural SDEs, learning proposals for stochastic sampling, and in particular diffusion models lays out nice grounding for future work.

Reproducibility: Yes

Additional Feedback: The abstract is a bit long and could probably be condensed, and would probably benefit from doing so. 27-28: 'toward target space the X' -> 'toward the target space X' 29: 'allowing to train them' -> 'allowing them to be trained' 29: 'ML' is already a well-known acronym for 'machine learning', and 'MLE' is generally better for 'maximum likelihood estimation'. 38, 41, 53, 61, 99, 106, 123, 131, 137, 162, 179, 195: Large space between paragraph title in bold and main text doesn't need to be there. It might also be worthwhile to separate the title from the paragraph text rather than joining them as in e.g. 38 & 41. 42-47: The true distribution of interest is denoted by muX, the distribution induced by the flow is pX, and the base distribution for the flow is muZ -- this is maybe confusing. Why not make the base distribution pZ? That is, pZ -> pX under F. 50-52: Although the slash is being used to distinguish between two different cases, it's ambiguous because the terms could also be interpreted as the ratio between two KL divergences, as well as the ratio between two densities. 50, 96: Specify exact sections in the supplementary material. 122-123: 'dynamics ... is given' -> 'dynamics ... are given' 197: 'log-likelihood of the test set' -> 'log-likelihood on the test set' Figure 1 and Figure 2 on two separate pages maybe? Figure 2 caption: 'An SNF transform' -> 'An SNF transforms' 217: 'data likelihood' -- again, the parameters of a statistical model have an associated likelihood function, not the data. 210, 213, 227/228: Again, there's really no reason for using a slash in each of these cases, and full sentences would read better. 195 & Table 3: 'VAE' isn't a method for enriching variational posteriors -- you've presumably compared various flows to a baseline diagonal Gaussian? -------------------------------------------- POST-REBUTTAL UPDATE -------------------------------------------- I'd like to thank to thank the authors for their response, and would like to see the paper accepted. On relating statistical physics to more classic ML: as you've promised, it would be nice to include a latent variable/variational bound interpretation (as Sohl-Dickstein et al 2015 'Deep Unsupervised Learning using Nonequilibrium Thermodynamics' do), and maybe also link to Deep Latent Gaussian Models (Rezende et al 2014 'Stochastic Backprop', Kingma et al, 'Autoencoding Variational Bayes').


Review 3

Summary and Contributions: This paper introduces a new variant on Normalizing Flows the authors name Stochastic Normalizing Flows. The general idea is to introduce stochastic operations in normalizing flows, while remaining able to compute importance weights, and thus remaining able to train these normalizing flows. This is done by considering importance weights taking into account the whole generative path (all intermediate random variables) rather than only the final output. Deterministic NF layers appear as a special case of this formulation. The method is quite general, and the paper focuses on applying it to the problem of sampling a probability distribution defined by an energy potential, linked to the general problem of molecular structure. In this context, the authors describe how Langevin dynamics can be used as a stochastic layer in the SNF context, and use them to iteratively morph the latent distribution (isotropic Gaussian) into the target probability defined by the energy function. Experimental results on synthetic data, and molecular data illustrate that SNF are significantly more expressive for the task of approximating an energy potential than regular NF are, and allow significantly faster convergence to the sought distribution than regular Metropolis MC, bringing together the best of both worlds. A final experiment shows that SNF can also be used to approximate the variational posterior of a VAE, with competitive performance.

Strengths: The authors provide solid theoretical grounding for their proposed method, and the well appendices illustrate its generality by deriving the relevant probability ratios for a wide range of stochastic transformations. The choice of the 2D densities as a synthetic example gives a good understanding of the relative behavior of SNF compared to Metropolis MC and NF, and the experimental results on the Alanine dipeptide test case seem quite promising (though I don't have a lot of background regarding molecular structure problems, so I cannot judge how representative of the problems of this field this particular example is). Overall, SNF appears to be a natural step forward regarding normalizing flows, and this paper seems to provide a solid base for it.

Weaknesses: This work is mostly theoretical-centered. I suspect to provide a very solid case regarding solving molecular structures (as is suggested to be the end goal by the introduction of the paper), more problems than just Alanine dipeptide would need to be used as benchmarked. Additionally, computing costs are not discussed. While the reader can get a general idea of how expensive SNF can be compared to NF by understanding how they work, I'd like to know in practice how this compares to pure-NF compared to training time, number of training iterations required for convergence, etc.

Correctness: I don't see any problem with the theoretical claims nor the empirical methodology, apart from the weaknesses reported earlier.

Clarity: The paper is quite clear and well written.

Relation to Prior Work: The prior work is clearly discussed and contextualized.

Reproducibility: Yes

Additional Feedback: The unnumbered equation l. 128-129 appears to have a typo, with a variable named x_t+1 while I suspect the intended name should be y_t+1 (as is written in the supplementary material). **POST-REBUTTAL RESPONSE** I'll thank the authors for clarifying the last missing points, and wish for this paper to be accepted. The parallel it provides between ML and non-equilibrium physics is very interesting!


Review 4

Summary and Contributions: The paper proposes a stochastic extension to normalizing flows where the transitions from one layer to another follow a proposal distribution. In such scenarios, exact marginalization is intractable. However, the authors propose to use importance weighting to estimate expectations w.r.t. such marginal distributions. Empirical evaluations further lend support to the proposed approach.

Strengths: + the problem pursued in the paper to sample from unnormalized distributions is challenging and fundamental to the neurips community + the solution to extend normalizing flows to their stochastic variants to improve expressivity of deterministic flows is also novel and principled + the exposition is clear for most parts

Weaknesses: - one of the motivations for stochastic NFs is that they overcome topological constraints. In the remainder of the paper, such constraints were not carefully formalized nor a reasoning provided for what makes NFs fail in such scenarios and how SNFs are particularly suited to fix these issues. Put differently, I wonder if the shortcomings of RNVPs in Figure 3 can be overcome with other architectures for normalizing flows, such as invertible resnets, iaf, maf etc. - the empirical evaluation in the context of existing works is largely restricted. For example, the schemes in [21, 35] could very well be applied here as well. Similarly, in the setup for MNIST/fashion datasets Table 3, I would have expected the default use implementation of flows for approximating intractable posteriors in a VAE is an inverse autoregressive flow (IAF) but instead the authors choose it to be a Real-NVP. - the unbiased guarantees for estimating expectations are only in the asymptotic limit. I would have been curious to see an analysis of the bias-variance tradeoff in the empirical evaluations. - related work such as A-NICE-MC [35] take special care to ensure that detailed balance is satisfied while using flows as proposals for MCMC. In the current work, it is unclear if such conditions are being satisfied in practice for SNFs.

Correctness: Yes, the claims and empirical methodology seem to be correct to the best of my knowledge.

Clarity: Yes, the paper is clearly written.

Relation to Prior Work: Some of the related methods eg, [33, 21, 35, 16, 15] have been briefly mentioned but it would be good to cover exact details for some key works in the appendix (especially those which can be case as a special instantiation of SNFs) as well as some empirical evaluation comparing these major variants with SNF as

Reproducibility: Yes

Additional Feedback: Post-rebuttal: Thanks for the rebuttal. I'd highly recommend including the clarifications in the final version as well. Also I got reminded of some more prior work that would be good to include in using importance weighting for correcting bias in expectations of generative models in a different, but related context. Bias Correction of Learned Generative Models using Likelihood-free Importance Weighting NeurIPS, 2019.

[Author Response · NeurIPS 2020]

1 We would like to thank all reviewers for constructive comments. Responses to criticism below, we will clarify all
2 unclear points in the final paper.

3 **R1, R2, R4**: Suggest comparisons with more advanced flow layers and trainable sampling techniques.

    1. We agree - we will add comparisons with more modern flow layers and trainable stochastic layers in the final
       paper as suggested by reviewers.

    2. General idea is: adding stochasticity increases the expressivity of a given flow architecture because stochastic
       layers do not have the restrictions coming from invertibility. For a given application problem, we can improve
       the SOTA flow method by adding stochasticity and turning it into a SNF.

    3. Some recent Monte Carlo methods utilize learnable proposals to improve the sampling efficiency, but they
       cannot yield exact reweighing scheme for tractable likelihoods like SNFs. But indeed, learnable proposals
       can be incorporated into SNFs, and we will discuss this and add comparisons in the final paper.

12 **R1, R3**: This work heavily relies on nonequilibrium statistical physics.

13 True, the motivation comes from Statistical Physics but it has important consequences for machine learning (ML). We
14 will provide a more easily accessible schematic of SNFs for ML readers and also give an interpretation of the loss
15 function as a variational bound in sampling path space.

16 **R1, R3**: Discuss computational cost / efficiency of SNFs versus NFs:

17 Agreed! We will provide a new Fig. 3f which shows KL divergence versus number of layers. Briefly, a stochastic
18 sampling step is a network layer, although a very simple one (in our implementation just point-wise operations without
19 parameters). We can show that even when just counting the number of layers, SNFs can surpass equally deep NFs.

20 **R4**: "Topological constraints" was not carefully formalized. Can the deficiencies in Fig. 3 overcome with other NF
21 architectures?

22 Formalization of topological constraints was done by others and we will discuss this more clearly in the final paper.
23 Yes, NF architectures have been developed to remedy these topological problems, but adding stochasticity can improve
24 them further (see first question+answer above).

25 **R4**: unbiased guarantees for estimating expectations are only in the asymptotic limit. Analyze the bias-variance
26 tradeoff for estimating empirical evaluations.

27 For the final paper, we will compare the errors of Table 1 for the biased case (uncorrected flow output distribution) and
28 the unbiased case (after applying Eq. 10) for different sample sizes – probably in a supplementary Figure.

29 As an explanation: our understanding is that the referee refers to the bias/variance tradeoff of invoking Eq. (10) on an
30 already trained estimator, which is somewhat different from the classical bias/variance tradeoff in statistical estimator
31 theory. A trained flow generates output distribution $p_X(\mathbf{x})$ that is different from target distribution $\mu_X(\mathbf{x})$, e.g. due
32 to limitations of its architecture, and will thus generate a biased estimate of an expectation value $\mathbb{E}_{\mathbf{x} \sim p_X}[O(\mathbf{x})]$. This
33 bias can be asymptotically removed by reweighting, e.g. Eq. (10), but this comes at the cost of a higher variance of
34 $\mathbb{E}_{\mathbf{x} \sim p_X}[O(\mathbf{x})]$ for a given number of samples. So we can compare the mean prediction error of $\mathbb{E}_{\mathbf{x} \sim p_X}[O(\mathbf{x})]$ before
35 and after applying Eq. (10). Note that we can counter the larger variance by drawing more samples, which is usually
36 preferred in physics applications, because unbiased estimators are essential there, and also taking samples from a
37 trained flow is fairly cheap compared to training time.

38 **R4**: Related work such as A-NICE-MC assumes that detailed balance is satisfied while using flows as proposals for
39 MCMC. It is unclear if such conditions are being satisfied in practice for SNFs.

40 Approaches like A-NICE-MC use detailed balance (DB) in each step, whereas SNFs rely on path-based detailed
41 balance between the prior and the target density. This is strictly more general, i.e. SNFs can use stochastic layers
42 that each respect DB (e.g., Metropolis Monte Carlo), but we can also give up DB in each step as long as the path
43 probability ratio (9) can be computed (e.g., Langevin dynamics). Will be clarified in final paper.

44 **R1**: Temperature used for the Alanine Dipeptide was 1000 K. Does the method fail with sharper modes?

45 1000 K was used in order to get a well-defined ground truth distribution as at 1000 K the MD simulation converges
46 quickly. Will be clarified in final paper. The method performs well with sharp modes – see Fig. 3.

47 **R4**: Explain how related methods, e.g. [33, 21, 35, 16, 15] are special instantiations of SNFs.

48 Excellent idea, will do.

49 **R1, R2, R3**: There were some typos, grammatical errors, undefined notations etc.

[Meta-Review · NeurIPS 2020]

There is a clear consensus among all reviewers that this is a strong paper which should be presented at the conference. Please do make sure to include all the clarifications from the rebuttal in the final camera-ready version of the paper, including the comparisons with more modern flow layers mentioned in point 1 in the response to R1, R2, and R4.